# Hybrid Dual-Polarization Synthetic Aperture Radar

**R. Keith Raney**

Applied Physics Laboratory, Johns Hopkins University, 1588 Keswick Place, Annapolis, MD 21401, USA;
k.raney@ieee.org

**Abstract:** Compact polarimetry for a synthetic aperture radar (SAR) system is reviewed. Compact polarimetry (CP) is intended to provide useful polarimetric image classifications while avoiding the disadvantages of space-based quadrature-polarimetric (quad-pol) SARs. Two CP approaches are briefly described, π/4 and circular. A third form, hybrid compact polarimetry (HCP) has emerged as the preferred embodiment of compact polarimetry. HCP transmits circular polarization and receives on two orthogonal linear polarizations. When seen through its associated data processing and image classification algorithms, HPC's heritage dates back to the Stokes parameters (1852), which are summarized and explained in plain language. Hybrid dual-polarimetric imaging radars were in the payloads of two lunar-orbiting satellites, India's Earth-observing RISAT-1, and Japan's ALOS-2. In lunar or planetary orbit, a satellite equipped with an HCP imaging radar delivers the same class of polarimetric information as Earth-based radar astronomy. In stark contrast to quad-pol, compact polarimetry is compatible with wide swath modes of a SAR, including ScanSAR. All operational modes of the SARs aboard Canada's three-satellite Radarsat Constellation Mission (RCM) are hybrid dual-polarimetric. Image classification methodologies for HCP data are reviewed, two of which introduce errors for reasons explained. Their use is discouraged. An alternative and recommended group of methodologies yields reliable results, illustrated by polarimetrically classified images. A survey over numerous quantitative studies demonstrates HCP polarimetric classification effectiveness. The results verify that the performance accuracy of the HCP architecture is comparable to the accuracy delivered by a quadrature-polarized SAR. Four appendices are included covering related topics, including comments on inflight calibration of an HCP radar.

**Keywords:** radar astronomy; synthetic aperture radar (SAR); compact polarimetry; Stokes parameters; Lunar Reconnaissance Orbiter; Earth observation; SAR applications; calibration

---

## 1. Introduction

This paper provides an overview of compact polarimetry, an imaging mode of synthetic aperture radar (SAR) that is an alternative to quadrature polarimetry (Appendix A). Compact polarimetry has proven to be appealing to a wide variety of users because it provides quantitative polarimetric-based classifications of a scene observed by a space-based imaging radar without being encumbered by the coverage limitations and technical complexities inherent to quadrature polarimetry. This review is designed to be accessible and informative for interested non-experts, while providing theoretical background and technical perspectives pertinent to radar experts, data processing specialists, and operational agencies.

Since its introduction to the SAR community nearly two decades ago, compact polarimetry has been a controversial subject. Partisan advocates defended quad-pol with a near-religious passion. They viewed those who promoted compact polarimetry—or "partial polarimetry" in their preferred jargon—as apostate, whose ideas were deemed to be sacrilegious, even dangerous. In the early days, loud voices of criticism often were raised in response to compact pol presentations at professional

conferences. One theme of those objections was that compact-pol could offer only an unacceptably poor alternative to quad-pol. In particular, it was argued that a compact-pol SAR would produce polarimetric classifications of vastly inferior accuracy, reliability, and repeatability relative to the high-quality classifications routinely achieved from airborne quad-pol systems.

Times have changed. There now is extensive experience with operational systems, and a body of comparative performance evaluations. Compact-pol orbital systems have been flown, from which data and their associated classifications have been completed. The two lunar compact polarimetric imaging radars—aboard NASA's Lunar Reconnaissance Orbiter (LRO) [1] and India's Chandrayaan-1 [2]—gathered all data in the hybrid dual-pol (HCP) mode. India's Earth-observing SAR on RISAT-1 [3] included an operational HCP mode. Japan's ALOS-2 imaging radar [4] included an experimental compact polarimetry mode (which has had limited use due in part to calibration issues).

Section 2 provides a summary of the compact-pol approach starting with the Stokes parameters which provide the theoretical foundations of compact polarimetry, and introduces the general idea and the early history of compact polarimetry for synthetic aperture radar imaging systems. Section 3 looks more deeply into a particular version of compact polarimetry, the hybrid dual-polarimetric mode. That sets the stage for commentary on the two orbital systems whose imaging radars first used that mode for polarimetric observations of the Moon. Section 4 focuses on Earth applications, with an emphasis on the question of performance quality, addressed through quantitative comparisons between compact-pol classifications and quad-pol decompositions for a variety of applications, and operational results from the RISAT-1 HCP mode. The answer is convincing: polarimetric classifications from hybrid dual-pol SAR data are comparable to those from a quad-pol SAR. The section also provides commentary on three groups of classification methods—two of which are error-prone, and a third which is recommended—for making such comparisons. The conclusions in Section 5 close the main body of the review.

Four Appendices are included. Appendix A summarizes quad-pol SAR and its advantages and disadvantages. Appendix B offers reflections on the principal finding that hybrid dual-pol and quad-pol classifications achieve very similar results. That finding raises two issues: why should that be true, and what are its implications. Appendix C looks at several ancillary but relevant topics under the heading of terminology. Appendix D suggests a method for the in-flight polarimetric calibration of a hybrid dual-pol SAR

## 2. Provenance

From a theoretical standpoint, one could argue that the foundational concept enabling compact polarimetry was discovered in 1852, as the key idea was first published by George Gabriel Stokes [5]. All that remained was for radar to be invented, space-based remote sensing radar systems to become commonplace, and a sophisticated user base to emerge. Section 2.1 introduces the Stokes parameters. Section 2.2 is comprised of a plain-language commentary on the principal characteristics of those parameters to illuminate their meaning and potential utility in the context of a compact polarimetric SAR. Section 2.3 recaps the early history of compact polarimetry pertinent to the Earth-observing SAR community.

### 2.1. Theoretical Foundation

The theory behind compact polarimetry is well-known [6,7]. A quasi-monochromatic partially-polarized electromagnetic field (EM) is represented by the ellipse swept out by its electric potential vector. Stokes showed that such a field could be represented by four numbers whose values could be determined from real observable quantities. Given the limited tools then available for observations of an optical field, his results were a major contribution. Although originated in the discipline of classical optics, Maxwell's elegant wave equations [8] apply to a wide span of frequencies, hence Stokes' parametric representation of an EM field are applicable to microwave systems including especially radar.

There are four essential themes embedded in Stokes' fundamental formulation: (*i*) observational data are required, collected through two channels each constrained to a polarization that is orthogonal to that of the other channel; (*ii*) relative phase is required in addition to the amplitudes of the dual-polarized images, which implies that signals through the two channels must be mutually coherent; (*iii*) four parameters—the Stokes parameters derived from the data—capture all of the polarimetric information conveyed in the observed field; and (*iv*) for any given EM field, the values of the Stokes parameters are independent of the particular basis vectors in which the coherent dual-polarized measurements are made.

The backscatter generated by a microwave radar illuminating the Earth's or a planet's surface is quasi-monochromatic and partially polarized, hence a Stokes parameter representation of the received data is appropriate and contains all of the available polarimetric information.

### 2.2. Stokes Parameters

The data resulting from a backscattered EM field as observed in the outputs of the pair of orthogonally-polarized receivers may be represented as a 2 × 2 coherence matrix or covariance matrix [9]. Those data are comprised of signal amplitudes, and the cross-products of their (complex) amplitudes. Each of the Stokes parameters (see Appendix C.2) is an elementary combination of two numbers drawn from the elements of one of those matrices. The resulting set of four real numbers, when evaluated at each pixel location in the multi-look SAR image domain, comprises a user-friendly form of the fundamental output data. Stokes parameter-based analysis methodology has a long and deep heritage in many fields (e.g., quantum physics, meteorology, crystallography, and radio astronomy).

Each of the Stokes parameters is an averaged quantity. The first Stokes parameter indicates the total intensity of the radar's backscattered field, which is the sum of the output powers of the two orthogonally-polarized receivers. That total power includes the radar's additive noise along with the polarized and randomly polarized constituents of the backscattered signal.

The other three parameters describe the properties of the polarized portion of the field. They have no additive noise component. The value of one of those Stokes parameters is the difference between the output powers of the two receive channels, so the averaged additive noise in the two channels cancels out. The other two Stokes parameters are derived from complex cross-products of the receivers' outputs, for which averaging also cancels out the additive noise contributions. In sum, additive receiver noise is random (modelled as a Gaussian process). Since it is not polarized, it cannot be a constituent in the three Stokes parameters that describe the polarized portion of the observed EM field.

No matter the receivers' polarization basis, the fourth parameter is the key to the ellipticity of the backscattered field, including the degree of ellipticity (ranging from linear to perfect circularity), and the rotational direction of its polarization vector (left or right). The second and third parameters together describe the degree of linearity of the polarized portion of the field, and the orientation of the dominant linear component. The degree of polarization is defined by the square root of the sum of the last three of the parameters each squared, divided by the first Stokes parameter.

### 2.3. Historical Overview

Within the discipline of polarimetric SAR applications, there were several influential early papers on compact polarimetry. The high-level objective of that line of inquiry was to exploit coherent dual-polarized radar data to realize many of the benefits of a quadrature-polarized system without its attendant disadvantages (see Appendix A). Compact polarimetry was proposed as a reasonable strategy when system resources (e.g., power, mass, data volume, and cost) or user demands (e.g., coverage, wide swath, or ScanSAR [10]) precluded full polarimetry. Potential users liked the idea from the outset, but many professionals in the technical radar remote sensing community were quick to find fault.

One of the first compact polarimetric concepts to appear is the $\pi/4$ mode [11], which posits radiating a linearly polarized field at 45° (with respect to the customary H or V orientation), then receiving the resulting H and V backscatter components coherently. Meteorological polarimetric radars

enjoy a multi-decade heritage with this mode [12,13], a discipline in which the π/4 mode is known as the slant mode. All modern weather radars routinely employ compact polarimetry.

The π/4 mode idea subsequently was extended to include circularly polarized transmission [14], with the suggestion that the two types of transmitted polarization should lead to equivalent results. It is now accepted that these two transmit polarizations lead to different results [15,16]. The most obvious reason for that difference is that slant-polarized polarization remains a single linear polarization. Yes, backscatter can be collected in the H- and V-polarized orientations, but at only half power. That loss cannot be recovered. Like-wise, the backscatter polarization constituent, which is orthogonal to the transmitted linear polarization is not excited.

Another form of compact polarimetry is to transmit a circularly-polarized field while receiving coherent dual-circular polarizations in response [17]. End-to-end circular polarization has an extensive heritage in Earth-based radar astronomy [18–20] from which very nice results have been obtained, including extensive precedent for backscatter classifications based on the Stokes parameters [20–22]. In principle, a space-borne SAR circularly polarized on transmit and dual-circularly polarized on receive would be a viable approach to compact polarimetry. However, such an approach implies additional electronics and hardware, and more stringent constraints on the system to facilitate calibration. Space-based implementation is better served by a simpler approach, having less demanding requirements.

The hybrid dual-polarized architecture form of compact polarimetry featured in this review satisfies both science and hardware requirements on a spacecraft embodiment. In this hybrid form, circular polarization is transmitted, while orthogonal linear polarizations are received. That architecture avoids the principal undesirable science features of the slant polarization, while also avoiding the hardware implications of an end-to-end circularly polarized radar.

## 3. Hybrid Dual-Polarization

The hybrid dual-polarization version of compact polarimetry [23] has emerged as the preferred method for operational systems [24,25]. An HCP SAR transmits a circularly polarized field (either L or R) and receives on two orthogonal linear polarizations (nominally H and V).

### 3.1. Rationale

From a user's perspective, why should circular polarization be a candidate for transmission? The answer follows from the higher-level objective, which is to generate an unbiased polarimetric characterization of an observed scene [23]. Transmission of any field that is dominated by a linearly-polarized component introduces rotational selectivity into the observation. This is true regardless of the degree of linearity, be it elliptical or purely linear, or its orientation, be it H or V or somewhere in between. When illuminated by linearly-polarized radiation, the backscattered polarization from a dihedral, for example, will depend to first order on the relative alignment of the axis of the dihedral with respect to the incoming field. In short, transmission of any field that has a dominant linearly-polarized component will lead to errors or omissions when attempting to classify double-bounce backscatterers [7]. The only globally valid option is to transmit circular polarization. A circularly-polarized transmitted field leads to rotationally invariant backscatter, since there is no favored linear alignment between the illumination and the scene's geometrical structure.

A CP radar by definition is constrained to only one transmitted polarization. Since transmitting only one linear polarization is not acceptable, a reasonable challenge by the user would be to ask that the radar transmit H and V simultaneously, which if it could be done would assure equal opportunity for H and V like- and cross-polarized backscatter constituents to be observed. If H and V were transmitted simultaneously having zero relative phase, the result would the slant (π/4) mode, hence linearly-polarized, still not acceptable. An innovative alternative is to transmit H and V simultaneously at 90° out of phase with respect to each other. This generates a circularly polarized field, but if and only if the H and V constituents have the same amplitude and waveform as well as

being orthogonal in phase. (From a science point of view, it does not matter which sense of circular polarization, L or R, is transmitted. That choice is best made during pre-launch testing of the radar by the instrument technicians, who should then select the one which shows the higher quality of transmitted polarization circularity.)

Circular transmit polarization enjoys a long and strong heritage in radar astronomy, where in that discipline selection of circular polarization originally was driven in part by the need to overcome Faraday polarimetric rotation of EM waves as they propagate through the ionosphere. That choice also was consistent with well-established traditions in radio astronomy [19].

If the transmitted field is circularly polarized, does the receiver also have to be? The answer to this question was anticipated by Stokes in 1852 [5,9,19]. Any pair of orthogonally polarized receiver channels would be acceptable, since the objective data product—the four Stokes parameters—have values that do not depend on the polarization basis in which the data are observed.

It turns out that a radar that transmits circular polarization is simpler to realize, its performance more reliable, and its calibration more robust, if the basis of the receiver channel pair is chosen to be orthogonal linear polarizations, such as H and V [23]. The principal reason is that neither receive channel has a like- or cross-polarized relationship with respect to the circularly-polarized transmissions. That choice has helpful hardware consequences, as it simplifies antenna and receiver implementations. Further, the mean signal levels in the two channels are always comparable, since neither receive channel is cross-polarized with respect to the transmitted polarization. That simple but essential fact is helpful because, in the usual end-to-end H and V quad-pol configuration [26], the mean signal levels through the two channels differ by 6–10 dB, which imposes additional requirements and tighter constraints on system design and calibration.

### 3.2. Hybrid Compact Polarimetry Goes to the Moon

The origin of an HCP SAR was motivated in large part by the requirements on two imaging radars that were being planned to orbit the Moon [24]. Their science requirements included measurement of the circular-polarization ratio, maximized potential to distinguish between backscatter types, and robustness to randomly oriented dihedral backscatter distributions. The implementation requirements included minimal mass and power. The hybrid dual-polarized architecture emerged as the optimum [27]. Circular polarization was transmitted, while the receivers were linearly polarized, nominally H and V. The baseline data product was stipulated to be the four Stokes parameters. This radar concept as implemented became the Mini-SAR on India's Chandraayan-1 and the Mini-RF on NASA's LRO, each independently launched in 2008, and successfully inserted into near-polar low-altitude lunar orbits.

These two SARs were the first HCP orbital systems [27], providing the same class of polarimetric imaging radar information from lunar orbit as Earth-based radar astronomy [28]. All data were collected in the hybrid-polarimetric mode, over 400 h in the case of LRO. The instruments each had mass less than 15 kg, antenna areas of about 1 m$^2$, and modest power and spacecraft accommodation requirements.

### 3.3. HCP Lunar Data Analysis

The values of the Stokes and their derived child parameters provide objective characterizations of geophysical properties of a lunar or planetary surface [19,20,29]. Along with the child parameters, a two-parameter set of classification variables (see Appendix C.3) was selected by the Mini-RF science team. The degree of polarization $m$ was the first, which has long been recognized as the single most important parameter characteristic of a partially polarized EM field [30]. The degree of depolarization $(1-m)$ is indicative of randomly polarized backscatter, typically arising from radar-quasi-transparent volumetric materials, such as lunar regolith. (The close relationship between entropy and degree of depolarization has been verified experimentally [31].) The mean value of $m$ for lunar radar data (at S-band) is about 0.6; on average about 40% of the radar backscatter from the Moon is randomly polarized.

The Stokes parameters offer several candidates for a second classification variable. Of these, the Poincaré ellipticity parameter *chi* ($\chi$) [32] is the most robust choice. It is one of the four principal components ($I$, $m$, $\chi$, $\psi$) that are necessary and sufficient to describe the polarized portion of a partially polarized quasi-monochromatic EM field of total intensity $I$. Further, the sign of $\sin\chi$ is an unambiguous indicator of even vs. odd bounce backscatter, thus directly responsive to the driving science requirements. The sign of $\chi$ indicates rotation sense even when the radiated EM field is not perfectly circularly polarized [33], as was the case with the Mini-RF and Mini-SAR radars. The resulting *m-chi* classification method has proven to be effective and reliable.

Discrimination between single (in general, odd) or double (in general, even) bounces (radar reflections from within an illuminated pixel within the scene) is known as a key to discovering lunar ice water deposits [34]. The required single-bounce vs. double-bounce characterization is expressed through the sign of the fourth Stokes parameter. $\sin2\chi$ (which is known formally as the *degree of circularity*) may be found from the Stokes parameters (see Appendix C.2) by $\sin2\chi = -S_4/mS_1$. (*Alert*: The sign, negative in this expression, is dependent upon the L or R handedness of the transmitted polarization and the data's coordinate convention as well single vs. double bounce discrimination.)

Three-color assignments for polarimetric imagery portrayal found in the literature usually use a combination of primary colors. The assignment of those colors to the three principal backscatter classes—random, single bounce, and double bounce—is at the discretion of the user.

An example of an LRO Mini-RF imaging product of the lunar surface classified through the *m-chi* method is shown in Figure 1. The Mini-RF science team adopted a color-coding backscatter classification based on three primary colors: green, blue, and red [24]. Green, indicating the randomly-polarized backscattered constituent, is proportional to (1-*m*). Blue usually was assigned to the single-bounce constituent, a class that includes Bragg scattering [35], a major constituent of lunar radar returns. In the coordinate system used for analysis, and respecting that the transmitted field was left-circularly polarized, this component is proportional to the square-root of ($1 - \sin2\chi$). Red, indicating double-bounce, is proportional to the square-root of ($1 + \sin2\chi$).

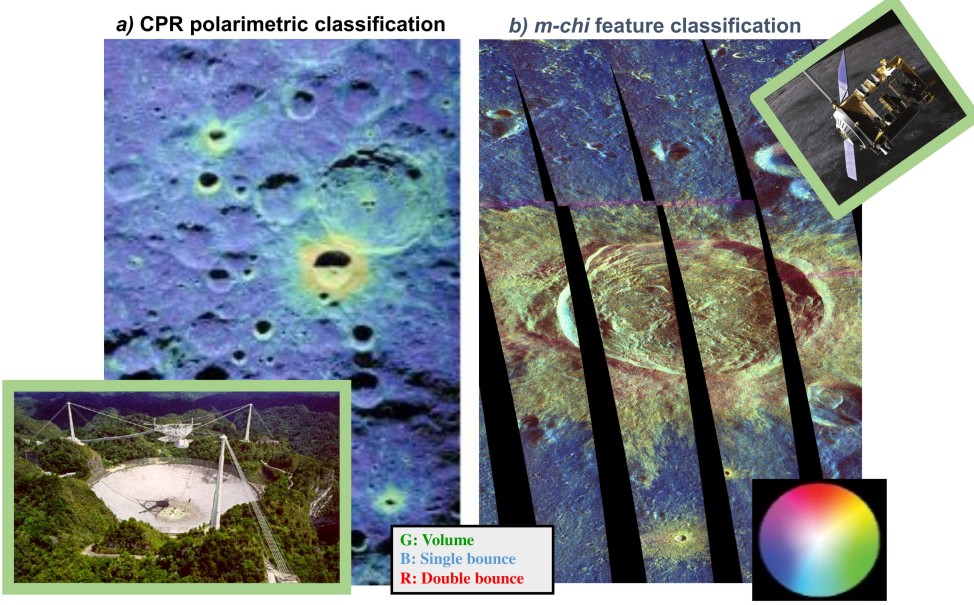

**a) CPR polarimetric classification**　　　**b) *m-chi* feature classification**

G: Volume
B: Single bounce
R: Double bounce

**Figure 1.** Examples of lunar impact crater polarimetric images as seen from the Arecibo Observatory (**a**) and an *m-chi* classification of hybrid dual-polarized data (**b**) from the Mini-RF radar aboard LRO (adapted from Figure 5 of [24]). The color wheel helps to retrieve meaning from the transition colors between primaries. In this example, yellow indicates dominant contributions from both random and double-bounce backscatter.

## 4. Performance in Earth Applications

How do classification results from HCP data compare to those from quad-pol data? One credible way to answer this question is to look at the results drawn from a variety of applications for which both QP and HCP data are available. There are now numerous peer-reviewed publications that report such comparisons, a number sufficient to provide an answer.

Many studies exploit an attractive feature of quad-pol SAR data to establish a nearly ideal comparison framework. Data from a QP radar can be transformed [36] from its conventional linearly-polarized format (H interleaved with V on transmit; H and V on each receive) into a different combination of transmit and receive polarizations (e.g., R on transmit; H and V on each receive). Hence, high-quality HCP data can be emulated (see Appendix C.4) from any given set of QP data. This approach assures that the ensuing comparisons of HCP vs. QP performance are objective and repeatable, since the same observational particulars apply for both scenarios.

What classification methodology is appropriate for HCP data? After reviewing [37] many dozens of candidate comparison studies that have appeared in journal articles, conference papers, or dissertations, three methodological approaches emerge that deserve comment.

### 4.1. Methodologies to Avoid

Recall that circularly-polarized transmissions (either R or L) are comprised of simultaneous H and V components, 90-degrees out of phase. In short-hand notation, the data observed through the H and V receiver channels could be described as (CH and CV) in general, or (RH and RV) in response to right-circularly polarized transmissions. It follows that in the post-detection radar image domain, following that notation the corresponding normalized backscatter coefficient $\sigma^0$ terms would appear as $\sigma^0_{CH}$ and $\sigma^0_{CV}$ (or $\sigma^0_{RH}$ and $\sigma^0_{RV}$ in the event that the transmission was R-circular). Such representations are misleading. A more transparent short-hand way to describe the data product from the H channel is (H + V) × H => (HH + VH). Similarly, data from the V channel are comprised of two terms (HV + VV). Respectively, the corresponding $\sigma^0$ terms are proportional to $|HH + VH|^2$ and $|HV + VV|^2$. In this expanded short-hand form, it is evident that the like- and cross-polarized contributions cannot be separated. CH and CV (or RH and RV) $\sigma^0$ terms are self-contaminated. This unwanted but unavoidable attribute compromises all studies whose hybrid dual-polarization classifications are based on $\sigma^0_{CH}$ and $\sigma^0_{CV}$ (or equivalently, $\sigma^0_{RH}$ and $\sigma^0_{RV}$). Advice would be: *Do not use this method*.

For those who are familiar with the analysis of quad-pol SAR data, a natural first step taken by such practitioners could be to expand HCP's native 2 × 2 covariance matrix into a 3 × 3 *pseudo-covariance matrix*. Such an expansion entails invoking certain symmetry and similarity arguments [14]. The resulting 3 × 3 matrix is a familiar form, hence investigators are able to apply standard quad-pol decomposition algorithms [38]. Unfortunately, that scheme does not turn out very well. Expanding a 2 × 2 matrix into a 3 × 3 matrix through the artifice of assumptions, no matter how clever, cannot add valid information. The 3 × 3 form may be more familiar, but at its best it is no better than its 2 × 2 foundation. At its worst, assumptions introduce biases that can lead to major classification errors [39]. There is no way for a user to know if or when to trust the results from the pseudo-covariance approach. Advice would be: *Do not use this method*.

### 4.2. Results from Appropriate Methodologies

The correct approach is simple in principle [37]. *Start* with HCP's 2 × 2 data matrix framework. *Stay* with those data. Do *not* expand that matrix. Do *not* make assumptions.

A variety of strategies are applicable to teasing out classifications from the 2 × 2 coherency (or covariance) matrix. The four Stokes parameters may be readily calculated from the 2 × 2 data. That is recommended as a starting point. Classifications based on the Stokes parameters enjoy a long and successful precedent, and consistently offer reliable results. One particular approach that seems to

work well in a variety of situations is the *m-chi* method [33,39]. The example in Figure 2 illustrates that the color-coded classified images from the same scene seen through an HCP or a QP polarimetric radar look almost identical.

**a)** Freeman-Durden QP decomposition　　　　　**b)** *m-chi* HCP feature classification

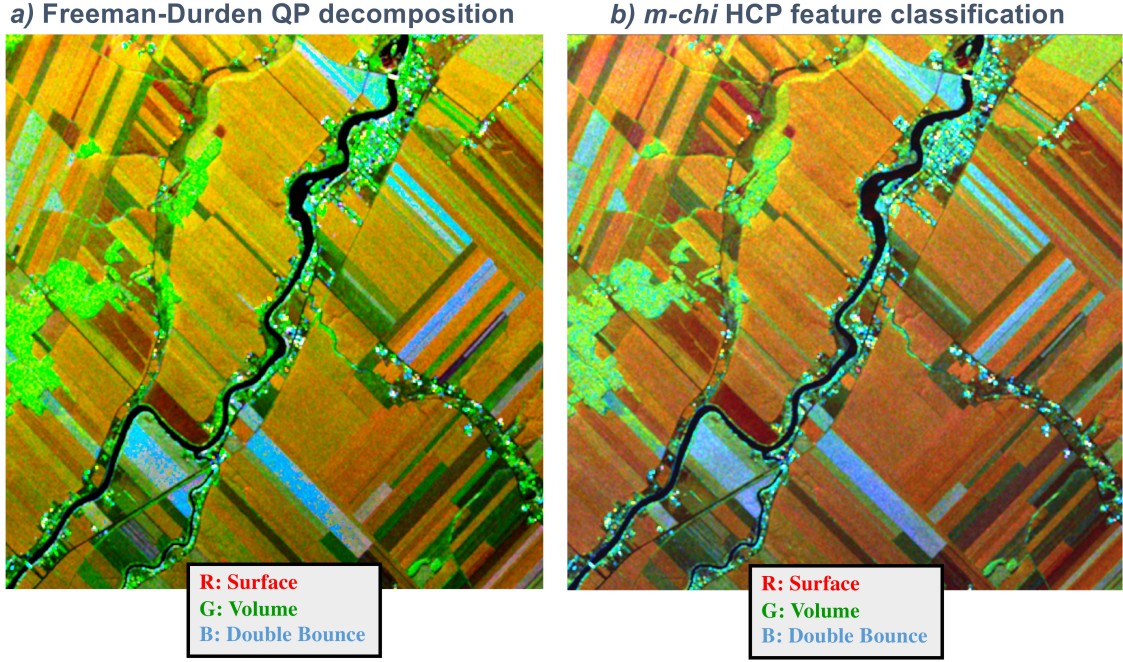

**Figure 2.** Example of classifications for a variety of agricultural fields as seen by (**a**) quad-pol and (**b**) hybrid dual-pol radars. Both images were derived from the same quad-pol data, using emulation to generate the corresponding HCP data (adapted from Figure 3 of [25]).

Other variations or methods may be used for classification, depending on the application, and the investigator's preferences. No one technique should be expected to be the best for all situations, although all such approaches should be based on their $2 \times 2$ data matrix origin. Acceptable alternative approaches include the following:

1.　Circularly-polarized like- and cross-polarized radar brightness constituents derived from the HCP data are sufficient for certain applications [40], but only if evaluated in the circularly-polarized domain (see Appendix C.2), for example $\sigma^0_{RL}$ and $\sigma^0_{RR}$.

2.　For decades, classical radar astronomy [41] has had success using the values of individual child parameters of the four-element Stokes vector, such as degree of polarization (*m*), circular-polarization ratio (*CPR*), degree of linear polarization ($m_L$), and sense of polarimetric rotation (see Appendix C.2) (opposite sense vs. same sense of rotation relative to that of the circularly transmitted EM field). These parameters as well as others may be useful in classification applications of Earth-oriented HCP radar data [42].

3.　The *m-chi* method draws upon two of the three classical Poincaré variables. The third, *psi*, indicates the orientation of the strongest linear polarization present in the backscattered field. This suggests that an *m-chi-psi* decomposition as a three-parameter classification scheme (which could be helpful in response to a transmitted field that is not perfectly circularly polarized).

4.　The *m-chi* method weights ellipticity by $(1 \pm \sin 2\chi)$ factors [24], where $-45° \le \chi \le +45°$. The *extrema* represent L or R circular polarizations. An alternative has been suggested [43] that substitutes the linearized factor $(1 \pm 4\chi/\pi)$. If greater sensitivity to ellipticity variations in the neighborhood of perfect circularity is of interest, then this version may be preferable.

5.　Although unsupervised classification techniques have been applied to polarimetric radar data for many years by several investigators, the approach seems neither to have attracted

much attention nor many adherents. The method deserves a second look. For example, multi-temporal agricultural crop studies using unsupervised classification of HCP-derived Stokes parameters achieved better results than quad-pol data classified through either entropy-alpha or Freeman–Durden decompositions [25].

To the surprise of many, classification accuracy by a hybrid compact-polarization architecture is proving to be very nearly as good as that from quadrature polarimetry. Studies using the appropriate methodology consistently yield very favorable HCP to QP comparisons. The authors of one investigation [44] concluded that the difference between quad-pol and hybrid dual-pol was "negligible" for their application, and those of another [45] found that the respective results were "comparable". That pattern is reinforced with similar endorsements: "1% to 3% differences between the HCP and FP classifications" [46]; "almost the same" [47]; "almost the same" [48]; and "equally well" [49]. A multi-disciplined investigation for a variety of Canadian applications [25], compiled in preparation for their RCM mission (whose main operational modes are all hybrid dual-polarimetric [50]), concluded that the classifications from hybrid dual-pol data differed by no more than a few percent with respect to those available from a quad-pol system. Guo et al. [51] found very good retrieval of rice biophysical parameters using a variety of classification methods and parameters on emulated HCP data, including one example for which $\sigma^0_{RH}$, a self-contaminated metric, turned out to be the best discriminator. (This result suggests that there may be applications in which one is interested in the total backscatter observed seen through one of the receiver's linearly-polarized channels.) Li and Perrie [52] found that HCP data supported sea ice classifications that were comparable to QP Pauli decompositions. Results were "similar" from emulated HCP and QP observations of ships and oil spills [53], and Dabboor et al. [54] concluded that CP and QP sea ice classification results were comparable over two test sites (96% vs. 99%, and 97% vs. 99%, respectively). Migliaccio et al. [55] showed that HCP does as well as QP for oil slick classifications, hence noting that HCP is the preferred polarimetric observation mode because of its wide coverage.

*4.3. Results from RISAT-1 HCP Data*

RISAT-1 (2012–2018) included HCP as an operational mode, the first Earth-orbiting mission to do so. RISAT-1 demonstrated that the HCP mode is compatible with ScanSAR [56], thus making available polarimetric imagery having swath widths of 223 km, a nearly ten-fold improvement over the 25-km width for RISAT-1's QP mode. A study using data from the HCP mode of RISAT-1 and the quad-pol mode of Radarsat-2 using near-contemporaneous coverage from the Greenland ice sheet [57] found the comparisons to be favorable. Several different land classification approaches included in studies by Kiran et al. using RISAT-1 data [58] yielded excellent informative results. Ponnurangam showed that, for soil moisture applications, HCP "works well" [59], and Uppala [60] found that compact polarimetry "works well" for maize crop discrimination. Espeseth et al. [61] achieved "similar results" when comparing classifications derived from RISAT-1 HCP data and HCP data emulated from Radarsat-2 for sea ice applications. Kumar et al. [62] applied unsupervised model-based classification techniques on the Stokes parameters from the RISAT-1 HCP mode to yield "very good" results.

**5. Conclusions**

The success of the HCP radars on two lunar missions is irrefutable, such that many proposals for imaging radars in response to opportunities for new missions to Venus or Mars have been based on the hybrid dual-polarimetric mode. India was the first to venture into Earth orbit with HCP, included in the suite of operating modes aboard RISAT-1. A decade ago Canada adopted HCP as its baseline architecture for all operational imaging modes aboard RCM, launched in June 2019.

During its introductory years within the SAR community, compact polarimetry was challenged vigorously by many professionals as being a poor substitute for quadrature polarimetry, and therefore undeserving of serious consideration. Given the acknowledged and demonstrated capabilities of QP SAR, the need for, let alone the anticipated benefits of, a leap to compact polarimetry was not obvious

at the time. Since then, that leap has enjoyed acceptance, largely due to users' interests. For users and operational applicational agencies, hybrid dual-polarity is emerging as the preferred form of SAR polarimetry because the alternatives are less attractive. Perhaps the most significant advantage for applications is that hybrid dual-polarimetric modes offer much wider coverage than the conventional quad-pol approach, at the same time supporting useful polarimetric classifications. For designers and system implementation and operations, an HCP mode implies technical demands on mission profile (including coverage, incidence, data burden, and power budget) that are less constraining than those for a QP system, thus enabling polarimetric SAR capabilities within realistic budget limitations as a routine operational mode.

The heart of the HCP mode resides in the fundamental formulation by Stokes in 1852 that four real parameters are sufficient to completely characterize the polarimetric properties of any quasi-monochromatic partially polarized EM field. Microwave imaging radars satisfy the initial conditions required for a Stokes parameter representation to be applicable. The success of the hybrid dual-polarized form of compact polarimetry is a direct consequence.

A dual-polarimetric radar's transmitted field's polarization must be the same, pulse to pulse, hence its polarization basis must be chosen thoughtfully. Circular polarization is the only option that gives rise to like- and cross-polarized backscatter constituents that are unbiased. Thanks to the Stokes parameter framework of the post-processing imagery, the polarization basis of the receiver can be selected based on hardware considerations. The optimum receiver basis for the receivers is orthogonal linearly polarizations when the transmitted field is circularly polarized.

Results from the lunar-orbiting HCP pioneers have been well received by the radar astronomy community. For Earth-observing applications, user's acceptance of hybrid compact-polarimetry now is wide-spread. Studies aimed at quantitatively comparing the classification performance of emulated HCP SAR data to QP decompositions for Earth applications consistently yield encouraging results. The first Earth-orbiting SAR to incorporate HCP—RISAT-1—provided data that justify the early optimistic predictions about compact polarimetric performance.

Evaluating the comparative performance of HCP vs. QP SARs is an ongoing enterprise. Such comparisons are to be encouraged, but they can be trusted only if the HCP radar's data are handled correctly. To do so requires that the $2 \times 2$ data array native to that architecture be the basis for classification methodologies. Expanding the HCP's $2 \times 2$ array into a $3 \times 3$ array induces classification errors. Reliance on image norms such as $\sigma^0_{RH}$ can lead to errors, because the output image elements from the H channel of an HCP SAR is actually $\sigma^0_{(HH+HV)}$, a form which makes obvious the fact that the cross-polarized power (HV) and the like-polarized constituent (HH) are bundled together, inseparable. It follows that an image made from the output of only one of the linearly-polarized channels of an HCP radar can be misleading.

The fact that results from HCP vs. QP classification comparisons are so favorable raises the obvious question: *why*? To formulate a credible response, the usual methods of QP classification deserve to be revisited. It could be true, for example, that the dominant polarimetric characteristics conveyed by the backscattered field observed by a QP radar are the same as those observed by an HCP radar. If true, then the sensitivity of a scene's signature to its illumination polarization has yet to be fully exploited through classification procedures that are adopted by operational agencies. As a review of the literature shows, rather few QP-based classifications work explicitly with the polarization signature, which as Zebker observed more than 30 years ago is one key to revealing a target's signature dependencies on the transmitted polarization. Without a polarimetric signature of an observed scene, it should be no surprise that results from the two architectures should be so similar.

The main point of this review is that QP as exploited in practice performs only marginally better than hybrid CP, with most examples showing near equivalence, and a few isolated examples showing HCP superiority. The reason is that QP decompositions do not take into full account the degree of freedom embedded in the QP $3 \times 3$ matrix data to search over all possible transmit polarizations to find one that best exposes a scene's characteristics of interest. Since wide area coverage by a polarimetric

SAR is a significant user's requirement, and classifications deduced from HCP data are to first order comparable to standard decompositions from QP data, hybrid compact polarimetry qualifies as an operational mode.

## 6. Patents

R. K. Raney, "Synthetic Aperture Radar Hybrid-Polarity Method and Architecture for Obtaining the Stokes Parameters of a Backscattered Field," U.S. Patent No. 7,746,267, July, 2010.

**Funding:** The early work behind this article was partially supported by the Lunar Reconnaissance Orbiter Project at the Johns Hopkins University Applied Physics Laboratory through contract with NASA under grant NNN06AA01C.

**Acknowledgments:** The author is indebted to numerous colleagues for their significant works and creative (and critical) thoughts and contributions over the years.

**Conflicts of Interest:** The author declares no conflict of interest. Other than occasionally asking probing questions, the funders had no role in the design of the study, in the writing of the manuscript, or in the decision to publish the results.

## Appendix A. Quadrature Polarimetry

Consider a SAR that uses orthogonal linear polarizations (such as H and V) on interleaved sequential transmissions, and then receives two orthogonal linear polarizations (such as H and V) after each transmission [26]. The resulting data may be organized into a $4 \times 4$ matrix. If those data are retained for subsequent operations, such a SAR is termed fully polarimetric. However, it is customary to reduce that array to a $3 \times 3$ matrix, by taking advantage of symmetries that are true in almost all known situations [63]. This form is known as quadrature polarimetry (QP). The simpler form is preferred, as its data files are significantly smaller, hence more efficiently incorporated into classification algorithms.

Since its introduction in the mid-1980s, quad-pol has been acknowledged as the "Rolls-Royce" of synthetic aperture radar (SAR) polarimetric imaging modes. In theory, that reputation is well-deserved, because the data collected by a QP SAR are necessary and sufficient to recover a complete characterization of the observed scene's polarimetric properties. The main advantage enjoyed by a quad-pol system is that the resulting Stokes parameter matrix can be obtained (either $4 \times 4$ or $3 \times 3$), thus capturing the complete polarization signature of the scene with an array of real numbers. Algebraic operations on those data can search over alternative transmit polarizations to select the one that best discriminates a scene's feature of interest, or to provide a graphic illustration of the variation of a target's response over all possible transmit polarizations. The result of such a search is expressed as a polarimetric signature [36]. With that in mind when considering polarimetric options, it is tempting to presume that quad-pol architecture should always be the favored choice.

Quadrature polarization for a space-based SAR comes at a significant cost, whose technical consequences are manifest in the radar system, which then imposes undesirable limitations on the resulting data products [64]. In particular, quad-pol requires that the radar's pulse-repetition frequency must be doubled, which implies that the imaged swath can be no more than half the width of a singly polarized or dual-polarized SAR. The average data rate should be comparable to that of a dual-polarimetric radar, although the data volume per pixel is doubled. The nearest range ambiguities (in conventional linearly polarized QP systems) are always cross-polarized with respect to the backscatter from the imaged swath. That is unfortunate, because range ambiguities—which are an unwanted nuisance—may be stronger than the desired signal, hence suppressing (or avoiding) them requires extra measures. For adequate signal-to-ambiguity data quality, it follows that the span of useful incident angles available from a conventional linearly-polarized QP SAR must be substantially smaller than for non-quad-pol systems. A QP radar requires twice the (average) transmitted power relative to that of its CP counterpart. Smaller swath width, limited choice in incidence, and twice the data volume per pixel are the most significant disadvantages for users.

## Appendix B. Quad-pol Reflections

The significance of the unsupervised classification attaining a result better than its quad-pol derived competition (Section 4.2) has an implication that goes beyond the cited agricultural application. It is proof by example that *conventional decompositions of quad-pol data are sub-optimum*.

A quad-pol SAR collects twice as much data per pixel than does a compact-pol system, which, being a dual-pol radar (hence inferior, as quad-pol advocates have asserted), has access only to the backscattered field, not the complete polarization signature of the scene. It would be tempting to presume that twice as much data collected should imply that classifications using those data should be twice as good. One takes for granted that access to more data about the observed scene should lead to much better polarimetric information retrieval. Evidently, that is not the case. Why, then, are the classification results from the two architectures so comparable?

There are three possible reasons: (*i*) the extra data contain no additional information; (*ii*) the usual quad-pol decomposition techniques do not take the extra data into account; or (*iii*) the extra data are accounted for, but they have very little influence on the classification.

The first candidate reason must be false. The second reason is likely to be at least partially true. The third reason is certainly true. For example, it is known that the data from a quad-pol SAR are necessary and sufficient to derive the polarimetric signature of the scene [36], which cannot be done with the data collected by a CP SAR. However, decomposition schemes in popular use do not include derivation of a scene's polarization signature.

One must conclude that most quad-pol decomposition results derive primarily only from polarimetric constituents conveyed by the field backscattered by the polarization actually transmitted. If so, then one should expect that there should be far more similarity between HCP and QP polarimetric measurements than is generally presumed, at least as seen through comparisons of their respective classification accuracies. Stated another way, given the state of the art, HCP classifications should be comparable to results derived from QP data, an expectation which is borne out by most well-done studies.

## Appendix C. Terminology

### Appendix C.1. Compact Polarimetric Radar Acronyms

Compact polarimetry in the literature often is represented by the abbreviation CP. In contrast, CP in other references is meant to convey one specific form, hybrid dual-polarimetry. Occasionally, one may find situations for which both meanings of CP are intended in the same source. *Alert*: There are references in which compact polarimetry (or CP) is meant—by assumption, default, or explicit definition—to apply exclusively to the pseudo-covariance matrix approach, which is error-prone (Section 4.1).

To indicate the hybrid version, some references use notations such as CTLR or CL—either of which is meant to imply circular transmit and linear receive, thus avoiding the implied ambiguity between compact pol in general, and hybrid dual-pol in particular. In this review, CP designates the family of compact polarimetric architectures, while HCP is reserved for the particular case of hybrid compact polarimetry. One reason for this suggested abbreviation rather than CTLR or CL is to discourage user reliance on self-contaminated variables (Section 4.1).

### Appendix C.2. Stokes Parameters

The Stokes parameters are an ordered set of four numbers. The names given those numbers vary, depending on the specific discipline or tradition preferred by their user. The three most common forms are $(I, Q, U, V)$, $(S_1, S_2, S_3, S_4)$, and $(S_0, S_1, S_2, S_3)$. In every case, in the set of four, the position of a given parameter indicates its meaning. The values of each of the four Stokes parameters does not depend on the polarization basis of their observation. However, the data elements needed to evaluate each

parameter does depend on the polarization basis of the observing receivers. For example, if the receiver channels are linearly polarized, then the second Stokes parameter is the difference between the received powers, whereas, if circularly polarized, that difference appears as the fourth Stokes parameter.

Note that the terminology "randomly polarized" indicates the portion of the field that is not polarized. That is preferred usage, because the word "depolarized" is meant in certain sources to indicate that portion of the backscattered field which is polarized but having an orientation that is orthogonal to that which was transmitted [18]. "Unpolarized" is an alternative, but still may be subject to ambiguous interpretation.

The Stokes parameters from an HCP radar can be used to emulate the output $\sigma^0$ image characterizations of a CP radar whose polarization bases are circular on both transmit and receive. In the field of radar astronomy, these terms are known as same-sense circular ($\sigma^0_{SC}$) and opposite-sense circular ($\sigma^0_{OC}$), where the terms *opposite* and *same* are reflective of the fact that the rotational sense of the transmitted circular polarization does not matter from a science perspective. Contrary to expectations based on experience with linearly polarized radar systems, the opposite-sense backscatter $\sigma^0_{OC}$ corresponds to "like pol" such as $\sigma^0_{HH}$. Single-bounce returns (and Bragg backscatter) usually are the strongest because the sense of polarimetric rotation is reversed by a single bounce. Conversely, the same sense backscatter $\sigma^0_{SC}$ corresponds to "cross pol" such as $\sigma^0_{HV}$. The OC constituents are almost always stronger than their counterparts, the SC data. The circular-polarization ratio CPR = SC/OC which usually is less than unity, becomes "interesting" in areas for which it is greater than unity, as observed with backscatter from planetary water-ice volumetric deposits, for example.

*Appendix C.3. Decomposition*

Analysis of quad-pol (or fully polarized) data requires retrieving information from their native $3 \times 3$ (or $4 \times 4$) matrices. Data interpretation in those scenarios implies determining a small number of parameters that together sufficiently characterize the salient properties of the data. Following in the footsteps of classical analysis, that process is known as decomposition [38,65], which provides an elegant mathematical formulation of that quest in the quad-pol SAR context. The term "decomposition" has carried over into the CP world. Given the simpler $2 \times 2$ matrix foundation (together with its Stokes parameter representation) of compact polarization data, the term "decomposition" is not necessary. The formulation by Stokes of the EM field in 1852, and its further illumination by Poincaré through spherical coordinates in 1892, provide eminently effective image classification tags for HCP data.

*Appendix C.4. Simulation vs. Emulation*

Quad-pol data from a known polarization configuration can be transformed by matrix operations on the original data [36] to represent an alternative radar architecture having any pair of transmit and receive polarizations. In particular, one may transform data collected by a conventional QP radar (e.g., Transmit H interleaved alternately with V; Receive H and V after every transmission) into data that would have been collected by an HCP radar (e.g., Transmit R; Receive H and V after every transmission). In the compact polarimetric literature, the dataset resulting from this transformation frequently is named a "simulation". That word has unfortunate connotations, such as "sham" or "counterfeit" (Merriam-Webster dictionary) along with "similar". Its use is discouraged. *Emulation* is a more appropriate word because the data resulting from the correct transformation yield exactly the same data as would have been produced by an HCP radar observing the same scene under the same conditions.

**Appendix D. Calibration of an HCP Radar**

The polarimetric calibration of an HCP radar is straight-forward, at least in principle. In brief, start with the receiver, and then work on the transmitter.

The objective for a well-calibrated receiver is to assure that the data at the outputs of the two polarimetric channels have the same phase and amplitude relationship as the incoming signals that

appear at the antenna. In the following summary description, the terminology "channel" is meant to include the antenna, receiver, and all other aspects of the signal paths, including waveguides, cables and connectors. After calibration corrections are applied, the two channels should be matched, in the sense that end-to-end the differential phase shift is zero, and their respective gains are equal.

The result expected from amplitude and phase calibrations of the HCP radar's receive channels is to have an instrument, after calibration, that preserves the polarimetric properties of the signals incident on the antenna so that the data products convey information about the observed scene, free from biases or distortions imposed by the instrument. Note that reliance on an external source of circularly-polarized illumination of the radar's receiving antennas is essential. The radar's own transmitted signal, either reflected by a corner reflector or replicated by a calibration transponder, cannot suffice because its imperfections will obscure measurement of the receive channel's gain and phase characteristics.

Before calibration, the relative gains of the two channels may not be equal. Likewise, before calibration, there is likely to be a differential phase shift between the two channels. The task is to measure the differential gain and phase shift between channels, and then to apply corrections during the post-reception processing such that the two channels after applying calibration corrections are matched.

The recommended method is to rely on an external reference. In the field of radio astronomy, for example, calibration strategies use a "candle", jargon for the radiation from a distant stable source [66] such as a quasar. The same approach is applicable to an HCP radar, for which an active radar calibrator (ARC) should illuminate the HCP radar's antenna with a near-perfect circularly-polarized field. A calibration dataset is generated each time the antenna pattern sweeps past the ARC. The lunar Mini-RF HCP radar was calibrated by this method using illumination from the Arecibo Observatory [27].

The antenna may have unwanted phase and amplitude variations in elevation and azimuth relative to its properties at boresight. In the azimuth coordinate, signal gain and phase off boresight can be observed directly in the Doppler domain of the image processor. Hence, if calibrations are needed to account for corrections that are a function of antenna azimuth angle, said variations can be directly measured, from which a calibration factor may be derived that includes Doppler-dependent corrections. Channel gain and phase characteristics also may be a function of the antenna's elevation pattern variations. In the normal side-looking attitude, the radar antenna's elevation angles correspond (on average) to radar range. Hence, range-dependent variations in antenna characteristics may be estimated by repeated passes over the ARC, for each one of which the spacecraft has a small known roll angle.

Once the receiver is calibrated, then it can be used to characterize the transmitted field. The recommended method is to rely on the backscatter from a known point target, such as a corner reflector (or transponder). The transmitted field is not likely to be perfectly polarized. Following a skilled design and build, its properties near boresight should be good. Beyond that, there may be departures from pure circularity in azimuth and in elevation. When the receive channels already have been calibrated to be nearly perfect, the transmitted field can be characterized. A multi-pass flight plan over a single corner reflector, or an array of corner reflectors distributed across track, will support estimation of range-dependent variations. Doppler-domain observations will provide azimuth-dependent information.

Unlike the receiver, whose imperfections should be substantially suppressed (ideally, eliminated) in the post-calibration dataset, the transmitted field cannot be corrected. It is what it is. If its imperfections are relatively small, then they should have minimal impact on the classifications to be derived from the radar's data. On the other hand, if the imperfections are found to be more substantial, as would be the case if the illuminating field were highly elliptical, then mitigating strategies should be explored. For example, data classification tools could be devised to use the polarimetric properties of the actual transmitted field as the reference basis for corresponding analysis tools.

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
