# Peer review of "Hybrid Dual-Polarization Synthetic Aperture Radar"

_remotesensing, doi:10.3390/rs11131521_

Round 1

Reviewer 1 Report

In this manuscript, the Author review the benefits of compact polarimetry compared to standard quadrature polarimetry (widely used at present).

The manuscript is well-written and the Authors shows deep knowledge of this subject. 

This reviewer enjoys the paper -although maybe sometimes the redaction seems too colloquial -"line 48, "...are very happy"-, and far from truly scientific/technical literature, although it is correct and acceptable (a different style).

This reviewer has some constructive concerns:

- due to the paper is a review, please, provide it a more tutorial-like look. That is, add more figures (for instance to better explain the so-technical concepts related to polarization, specially in the appendices). Include also basic equations (show matrices!) and also include more results (classification ones),

-remark both the benefits of the compact polarimetry and the shortcomings (this can be includes around lines 416-->).

Some typos:

- some references appear with (for instance, [48],  "pp. 471-473), and others, (for instance [57], "172-183",

- reference [50] has the DOI but not the vol., pp....; revise.

Author Response

Reply to Reviewer #1

Comments and Suggestions for Authors

In this manuscript, the Author review the benefits of compact polarimetry compared to standard quadrature polarimetry (widely used at present).

    Actually, standard quadrature polarimetry is notwidely used at present. There has been wide-spread use of airborne QP SAR data, but only a relatively small amount of QP SAR data from space-based assets have been collected and classified, when compared against the very large quantities of data collected, analyzed, and put to operational work derived from standard and modes. With Radarsat-2 for example, the amount of ScanSAR data collected is orders of magnitude greater than that for QP SAR data. For example, Canadian Ice Center’s Arctic monitoring relies on conventional dual-pol data, not QP data. There has been no orbital SAR whose QP mode has been deemed “operational”, in the sense that it has become the mainstay of routine and repeated user applications. In short, QP SAR data is widely admired by theoreticians and used when available for experimental purposes, but it is seldom used by operational agencies.

The manuscript is well-written and the Authors shows deep knowledge of this subject. 

     Thank you.

This reviewer enjoys the paper -although maybe sometimes the redaction seems too colloquial -"line 48, "...are very happy"-, and far from truly scientific/technical literature, although it is correct and acceptable (a different style).

     This particular sentence has been deleted. Thoughts along the same line are incorporated into the conclusions section, hopefully at a more acceptable professional level.

This reviewer has some constructive concerns:

- due to the paper is a review, please, provide it a more tutorial-like look. That is, add more figures (for instance to better explain the so-technical concepts related to polarization, specially in the appendices). Include also basic equations (show matrices!) and also include more results (classification ones),

     The key observation in this critique is that the paper is a review. Correct. Put another way, it is not a tutorial. As a review, its intended audience is meant to span non-expert users as well as expert practitioners. The former would never read past the first equation, whereas the latter do not need yet another treatise weighed down by equations. I have written this review in the spirit of Stephen Hawking, who noted early in his “A Brief History of Time” that a book’s readership is reduced by a factor of two for every equation appearing in the work. In his case, he limited himself to one, E=mc2. In my case, I limit myself to one, but buried in the text, as it is nowhere as fundamental as Hawking’s inclusion.

    Figures? That question poses challenges. I have chosen to use only two illustrations, one drawn from the planetary (well, lunar actually) orbital CP pioneers, and the other from Earth observations (agriculture). Both are figures from teams with which I have had the privilege of working. To go beyond that, to have sufficient figures to illustrate the variety of polarimetric SAR applications that could benefit from hybrid CP coverage, would lead to a mini-dissertation on the subject. Even if only a few more illustrations were to be included, that could be construed as unfair to all those whose good works were not included. Instead, the review cites many sources whose pages include excellent examples. Readers should be motivated to seek out relevant illustrations pertinent to their particular application.

-remark both the benefits of the compact polarimetry and the shortcomings (this can be includes around lines 416-->).

     The text does include comments on the benefits and shortcomings of CP, the most dominant of which is its inability to search for an optimum transmit polarization when analyzing the scene’s backscatter characteristics. The background issue here is that those who are protagonists for QP start from the position that CP in any form, including hybrid compact polarimetry, must be inferior to QP. Given that initial attitude, then they expect that those shortcomings should be made clear by CP supporters. 

    The main point of this review is that QP as exploited in practice performs only marginally better than hybrid CP, with most examples showing near equivalence, while a few isolated examples show CP superiority. The reason is that QP decompositions do not take into full account the degree of freedom embedded in the QP 3x3 matrix data to search over all possible transmit polarizations to find one that best exposes a scene’s characteristics of interest. One could make the case that compact polarimetry’s greatest shortcoming is a decided lack of support from SAR practitioners whose point of view is shaped by QP’s elegant theoretical modelling, rather than an inherent weakness in polarimetric classifications derived through an HCP system.

Some typos:

- some references appear with (for instance, [48],  "pp. 471-473), and others, (for instance [57], "172-183",

- reference [50] has the DOI but not the vol., pp....; revise.

     The list of references is presented in a variety of styles. That is a “work in progress”.  According to instructions provided to this author by the Editor, their harmonization into the style preferred by Remote Sensingwill be done by the journal’s editorial staff once the paper is found to be acceptable by the reviewers.

Reviewer 2 Report

Dear Author,

thank you for submitting this review paper that could be very useful and interesting across the scientific community working on SAR compact-polarimetry. The manuscript is very well written and address in a robust and effective way most of the important questions arisen in years of research on the topic. It also proposes itself as a guideline for all the scientists intended to exploit SAR compact polarimetry. 

I have only some minor concerns that, as a reader working on the topic, may help to improve the manuscript. They are as follows:

- Please define all the acronyms at their first instance and only once;

- Together with actual Risat-1 and future RCM CP SAR missions, the operating Alos-2 mission equipped with 2 CP imaging modes, even if with calibration issues, should be at least mentioned.

- It would be beneficial to support subsection 4.1 with some showcases (with proper references) where discourage approaches for the exploitation of CP information were used and the effects of this.

- Lines 337-339, please rephrase the sentence.

- About section 4.2, it would be interesting and beneficial for the whole community, since CP classification performance is scenario/application-dependent, to present a literature showcase (e. g., one per domain as hydrosphere-cryosphere-land-urban) where differences between CP and QP classification results are pointed out and quantified.

- Please clarify point 5 in subsection 4.2.

- In my opinion, subsection 4.3 seems to be quite of context there and can be briefly included into the introduction.

- Lines 384-385, please rephrase the sentence.

- References about the exploitation of QP and CP polarization signatures should be added.

- Appendices: in my opinion, some part of the appendices reach a too much fine level of details for an expert reader, while a non-expert reader can be directed to some well-established textbook. In addition, some part of the appendices can be shortened and included into the main body of the text. Summarizing: A could be shortened; B could be included into the text; C could be shortened and included into the text; D could be dedicated to calibration of CP SAR only.

- Lines 486-489: the difference between item ii) and iv) is not so clear, please clarify.

- Lines 493-494: in literature there are many example on the exploitation of QP polarization signatures (fewer if considering CP SAR). Please check.

- Line 734, check Authors' spelling.

- References could be improved. Unify reference format. Reference to papers not published on peer-review international journals should be strongly motivated (or they must be replaced by corresponding full papers if any). Being a review paper, some interesting studies on CP SARs and their applications are missed and should be included: 10.1109/TGRS.2015.2494860,

10.14358/PERS.81.7.557, 10.1109/LGRS.2016.2595058, 10.1016/j.rse.2013.08.035,   

10.1109/JSTARS.2016.2584542, 10.1109/TGRS.2016.2574561, 10.3390/rs9111088,

10.1080/01431161.2016.1184353, 10.1109/JSTARS.2012.2182760, 10.3390/app7040368,

10.1109/JSTARS.2017.2768378, 10.1109/JSTARS.2014.2359141, 10.3390/rs10040594,

10.1109/TGRS.2013.2242479, 10.1109/LGRS.2011.2158983, 10.3390/rs9020168. Please consider those works only as a suggestion and not a limiting list, different references can be included.

Author Response

Reply to Reviewer #2

Comments and Suggestions for Authors

Dear Author,

thank you for submitting this review paper that could be very useful and interesting across the scientific community working on SAR compact-polarimetry. The manuscript is very well written and address in a robust and effective way most of the important questions arisen in years of research on the topic. It also proposes itself as a guideline for all the scientists intended to exploit SAR compact polarimetry. 

    Thank you for your constructive comments. Clearly you have understood the spirit and intended audience of this paper. I have added a sentence in the introduction to help others to see that, which seems not to be obvious to all readers, at least judging by the critique of another reviewer.

I have only some minor concerns that, as a reader working on the topic, may help to improve the manuscript. They are as follows:

- Please define all the acronyms at their first instance and only once;

     Done.

- Together with actual Risat-1 and future RCM CP SAR missions, the operating Alos-2 mission equipped with 2 CP imaging modes, even if with calibration issues, should be at least mentioned.

     Done. They all are mentioned, with the appropriate caveats “only mode”, “operational mode”, and “experimental mode”, defended with references. As you seem to appreciate, the RISAT-1 CP mode quickly became accepted as a valuable and popular operational mode, in contrast to the two  CP modes aboard ALOS-2 which did not meet that level of acceptance.

- It would be beneficial to support subsection 4.1 with some showcases (with proper references) where discourage approaches for the exploitation of CP information were used and the effects of this.

     At one level I agree. However my style is to avoid critical commentary on other’s work, at least by name and citation. I know and have worked with many of the individuals in question, such as Cloude, Pottier, Nord, J.-S. Lee,  Ainsworth, Souyris, Boerner, and several others. I have added one new citation (drawn from your suggested references) that defends the position that the pseudo-covariance matrix approach does not work in all situations, with evidence to illustrate the issues.

- Lines 337-339, please rephrase the sentence.

     This comment does not make clear what is to be accomplished. I have rephrased the closing thought in that passage, hopefully addressing your concerns.

- About section 4.2, it would be interesting and beneficial for the whole community, since CP classification performance is scenario/application-dependent, to present a literature showcase (e. g., one per domain as hydrosphere-cryosphere-land-urban) where differences between CP and QP classification results are pointed out and quantified.

     Section 4.2 has been amended (although only by a small amount), and a new section 4.3 has been added, in part to broaden the scope of applications, and also to illustrate specifically the excellent results that have followed from the hybrid CP mode on RISAT-1.

- Please clarify point 5 in subsection 4.2.

     The point made is that unsupervised classification, a technique often used with multi-spectral data, is not commonly used by SAR practitioners. The reference cited provides an excellent example of what could be accomplished. That stands on its own. In addition, one of the examples cited in the RISAT-1 paragraph uses unsupervised classification, again with very good results.

- In my opinion, subsection 4.3 seems to be quite of context there and can be briefly included into the introduction.

     I agree with the observation, but it has been moved to section 5 ( conclusions), and modified accordingly.

- Lines 384-385, please rephrase the sentence.

     The sentence remains as written. If further clarity (or interest) is warranted, the reference cited should suffice. This passage is not meant as a recommendation as such, but serves only to illustrate that there is room for a user’s creativity when trying to develop classification tools that work--within the “rules” explained earlier in the paper--for their particular applications.

- References about the exploitation of QP and CP polarization signatures should be added.

     There is a reference in context to QP polarization signatures, indeed, the prime reference. On the other hand, CP polarization signature tools, in the original sense of the word, are not possible since the transmitted field is a given, and cannot be manipulated to search for a “better” Tx choice. There have been efforts in that direction (e.g. Charbonneau), but in my view such offerings are a bit of a stretch. Further, the closing comments of this paper encourage further research and development on exactly this topic.

- Appendices: in my opinion, some part of the appendices reach a too much fine level of details for an expert reader, while a non-expert reader can be directed to some well-established textbook. In addition, some part of the appendices can be shortened and included into the main body of the text. Summarizing: A could be shortened; B could be included into the text; C could be shortened and included into the text; D could be dedicated to calibration of CP SAR only.

     In early drafts of this paper, most of the materials now appearing as appendices were originally in the main body of text. One by one, those materials were moved to their position as appendices, for the simple reason that when in the body of the paper, they detracted from the central theme of the hybrid CP story. The appendices have been written to provide insight, background, and potential ambiguity suppression for a non-expert, while also clarifying and (hopefully) standardizing concepts and nomenclature that should be noticed and of use by experts. It is experts who would be motivated to seek these subjects in references, not the non-experts. To reinforce this latter thought, I know of no “well-established textbook” that handles the matters covered in the appendices (e.g. the Stokes parameters) at a level appropriate for a non-expert reader such as a potentially interested user of polarimetrically classified imaging radar data products. 

- Lines 486-489: the difference between item ii) and iv) is not so clear, please clarify.

     Actually there is a difference, but at a subtle level. In response to your query, the two thoughts have been merged in the revision.

- Lines 493-494: in literature there are many example on the exploitation of QP polarization signatures (fewer if considering CP SAR). Please check.

      Whereas your comment is correct, it misses the point. The text has been modified in an attempt to make the point clearer. What is the point? The majority of polarimetric classifications evident in the literature, at least those that I have seen over the past 20 years and more, is based exclusively on “decomposition” techniques. Of those, most are two-parameter methods, including Freeman-Durden and Entropy-Alpha. (The latter in all cases known to me provides inferior classifications when compared to hybrid PC results.) Three-parameter decompositions such as Yamaguchi or Touzi are often better than the two-parameter methods, but still do not approach what should be the standard set by a complete polarimetric signature. Most discussions, at least from among those that I have seen, on polarimetric signatures, seem to be making the point “gee whiz, look what I can do”, rather than using the device to advantage in a particular classification application.

- Line 734, check Authors' spelling.

     Done. (The citation was in error in several regards. Thank you)

- References could be improved. Unify reference format. Reference to papers not published on peer-review international journals should be strongly motivated (or they must be replaced by corresponding full papers if any). 

     Thank you for your valuable list of suggested references. I have incorporated most of them into the revised work. Several are cited in the new section 4.3 devoted to RISAT-1 results, and two others elsewhere as appropriate. In only a few instances are papers from conferences and the like cited, but for them I have found no peer-reviewed source that includes the material of interest (e.g ScanSAR and ALOS-2 inclusion of CP only as an experimental mode). At least one other conference paper was included as a matter of documenting original precedence. 

    The format for the references was sent to reviewers in the form that I sent the paper to the Editor. Said Editor had assured me prior to my submission that their staff would harmonize the citation formats according to their preferred style once the paper was acceptable to reviewers. That pledge was in response to my query to the journal about certain details of their style, which as far as I have been able to discern does not have a name, nor is it internally consistent.

Round 2

Reviewer 1 Report

My concerns were properly attended by the Author. 

This reviewers agrees completely with the reply to my concern about to provide to the manuscript a more "tutorial-like" air: well replied!.